# Controlling contractile instabilities in the actomyosin cortex

**Masatoshi Nishikawa[1,2,3†], Sundar Ram Naganathan[1,2,3‡], Frank Jülicher[2], Stephan W Grill[1,2,3*]**

[1]Biotechnology Center, Technical University Dresden, Dresden, Germany; [2]Max Planck Institute for the Physics of Complex Systems, Dresden, Germany; [3]Max Planck Institute of Molecular Cell Biology and Genetics, Dresden, Germany

**Abstract** The actomyosin cell cortex is an active contractile material for driving cell- and tissue morphogenesis. The cortex has a tendency to form a pattern of myosin foci, which is a signature of potentially unstable behavior. How a system that is prone to such instabilities can rveliably drive morphogenesis remains an outstanding question. Here, we report that in the *Caenorhabditis elegans* zygote, feedback between active RhoA and myosin induces a contractile instability in the cortex. We discover that an independent RhoA pacemaking oscillator controls this instability, generating a pulsatory pattern of myosin foci and preventing the collapse of cortical material into a few dynamic contracting regions. Our work reveals how contractile instabilities that are natural to occur in mechanically active media can be biochemically controlled to robustly drive morphogenetic events.

**\*For correspondence:** stephan. grill@tu-dresden.de

**Present address:** [†]Department of Frontier Bioscience, Hosei University, Koganei, Japan; [‡]The Francis Crick Institute, London, United Kingdom

## Introduction

Alan Turing described in his seminal 1952 paper the ability of an initially homogeneous spatial system that contains diffusing and chemically interacting species to form a self-organized pattern (*Turing, 1952*). Turing's original conjecture was that such processes contribute to the patterning of developing organisms. While many examples have been found that are compatible with this idea (*Kondo and Asal, 1995*; *Müller et al., 2012*; *Sheth et al., 2012*; *Raspopovic et al., 2014*), self-organized patterning in morphogenesis, however, is known to not only rely on biochemical regulation but also depend on cell-and tissue scale active mechanical processes (*Turing, 1952*; *Howard et al., 2011*). General physical mechanisms by which the interplay between regulatory and mechanical processes endows active biological materials to form self-organized spatiotemporal patterns have remained largely unexplored.

Actomyosin contractility (*Bray and White, 1988*; *Salbreux et al., 2012*) is an essential cellular mechanical process, responsible for driving many cell- and tissue scale morphogenetic events (*Murrell et al., 2015*; *Levayer and Lecuit, 2012*). The cortex consists to a large extent of actin filaments and myosin motor proteins, forming a thin layer underneath the cell membrane that can be thought of as a thin film of an active gel (*Salbreux et al., 2012*; *Jülicher et al., 2007*). Contractility by myosin motor proteins generates active tension in the gel, and gradients in active tension are known to generate cortical flows of this layer (*Mayer et al., 2010*). Cortical flow participates in forming the cytokinetic furrow (*Bray and White, 1988*; *Benink et al., 2000*; *Yumura, 2001*; *Eggert et al., 2006*), and drives polarization of the one-cell stage *C. elegans* embryo (*Hird and White, 1993*; *Guo and Kemphues, 1996*; *Cheeks et al., 2004*; *Munro et al., 2004*; *Goehring et al., 2011*). Highly contractile cortices, like the one driving polarization in *C. elegans*, tend to exhibit transient accumulations of myosin that form a pulsatile pattern. Pulsatile actomyosin patterns are ubiquitous in development (*Munro et al., 2004*; *Martin et al., 2009*; *Solon et al.,*

*2009*; *Rauzi et al., 2010*; *Roh-Johnson et al., 2012*; *Maître et al., 2015*), and it has been suggested that they result from positive feedback and contractile instabilities (*Kruse and Jülicher, 2000*; *Bois et al., 2011*; *Gowrishankar et al., 2012*; *Kumar et al., 2014*; *Munjal et al., 2015*; *Hannezo et al., 2015*). A contractile instability causes the cortex to become inhomogeneous over space, with cortical material collapsing into contracting regions (*Bois et al., 2011*; *Alvarado et al., 2013*). Theoretical work has shown that contractile instabilities are inevitable when contractility is high enough (*Bois et al., 2011*), raising the question of how a system that is prone to such instabilities can reliably drive morphogenesis. Here we show that there indeed exists a contractile instability in the actomyosin cortical layer of the *C. elegans* zygote, and we discover that this instability is controlled by a RhoA oscillator.

## Results and discussion

In order to investigate spatiotemporal patterns in the *C. elegans* cortex, we first sought to see whether non-muscle myosin II (NMY-2) in the *C. elegans* zygote displays pulsatile dynamics (*Munro et al., 2004*). For this, we determined the temporal derivative of the average NMY-2 intensity (*Figure 1A*, averaging over a region in the posterior indicated by a white box), as a proxy of myosin foci assembly and disassembly behavior (*Figure 1A–C*). We also quantified the time-dependence of the average speed of cortical flow in this region as determined by Particle Image Velocimetry (PIV, see Appendix for detail) (*Figure 1B,C*). Notably, both quantities exhibited signs of oscillatory behavior (*Figure 1C*) and an auto-correlation analysis revealed periodic changes in both quantities with a time constant of approximately 30 s (*Figure 1D,E*). To conclude, the myosin foci pattern in the *C. elegans* zygote exhibits pulsatile, oscillatory dynamics.

We next sought to understand where this oscillatory behavior comes from. One possibility is that positive feedback mediated by RhoA (RHO-1 in *C. elegans*) (*Bement et al., 2015*), a key activator of myosin (*Jenkins et al., 2006*; *Motegi and Sugimoto, 2006*; *Schonegg and Hyman, 2006*), plays a role in generating this pulsatile pattern (*Munjal et al., 2015*). We investigated the dynamics of active RhoA by use of a GFP fused anillin homology domain (AHPH) probe, to image the GTP-bound active form of RhoA (*Tse et al., 2012*). We find that active RhoA forms a dynamic, pulsatile pattern that is similar to that of myosin, with both active RhoA and myosin co-localizing in pulsatile foci (*Figure 1F*, *Figure 1—figure supplement 1*, and *Video 1*). We speculated that flow-based transport of an activator of myosin could give rise to positive feedback and a contractile instability (*Bois et al., 2011*), favouring the spontaneous formation of self-organized pulsatory patterns (*Munjal et al., 2015*). However, testing for this possibility requires knowledge of the kinetics of active RhoA mediated myosin recruitment coupled with a hydrodynamic description of active cortical mechanics, for evaluating if the full mechanochemically coupled system indeed is unstable.

We set out to test if coupling RhoA mediated myosin recruitment to gel flow and advection results in an intrinsically unstable cortex (*Figure 2A*). To this end, we sought to determine the effective reaction kinetics of the regulatory interaction between active RhoA and myosin *in vivo*. We developed a method of measuring the kinetic diagram of active RhoA mediated myosin recruitment (CO-moving Mass Balance Imaging; COMBI): We investigated the mass balance of both species in the comoving frame of reference of the flowing cortex, under consideration of the effects of dilution/enrichment by divergent/convergent gel flow (*Figure 1G*) (*Vallotton et al., 2004*). In the frame of reference of the embryo, concentrations of myosin and active RhoA can change due to transport by flow (advective fluxes) or due to association/dissociation (chemical fluxes). The chemical fluxes $R_r$ and $R_m$, where r denote active RhoA and m denotes myosin, correspond to reaction terms that capture turnover and biochemically regulated recruitment effects. They can depend on the concentrations of both species.

COMBI determines the average changes of per area myosin concentration ($c_m$) and of active RhoA concentration ($c_r$) due to turnover/regulation and as a function of the concentrations of both species. This provides us with information of the reaction kinetics of RhoA $\left(R_r(c_r, c_m)\right)$ and myosin $\left(R_m(c_r, c_m)\right)$ in the myosin and active RhoA concentration phase space (*Figure 1G*). We determined $c_m$ and $c_r$ every 5 s by spinning-disk confocal microscopy (Materials and methods). Advective fluxes account for the effects of dilution/enrichment by divergent/convergent gel flow, and these were determined by measuring the velocity field of cortical flow by particle image velocimetry (PIV), an image-based crosscorrelation analysis (*Mayer et al., 2010*; *Raffel et al., 2007*) that quantifies the

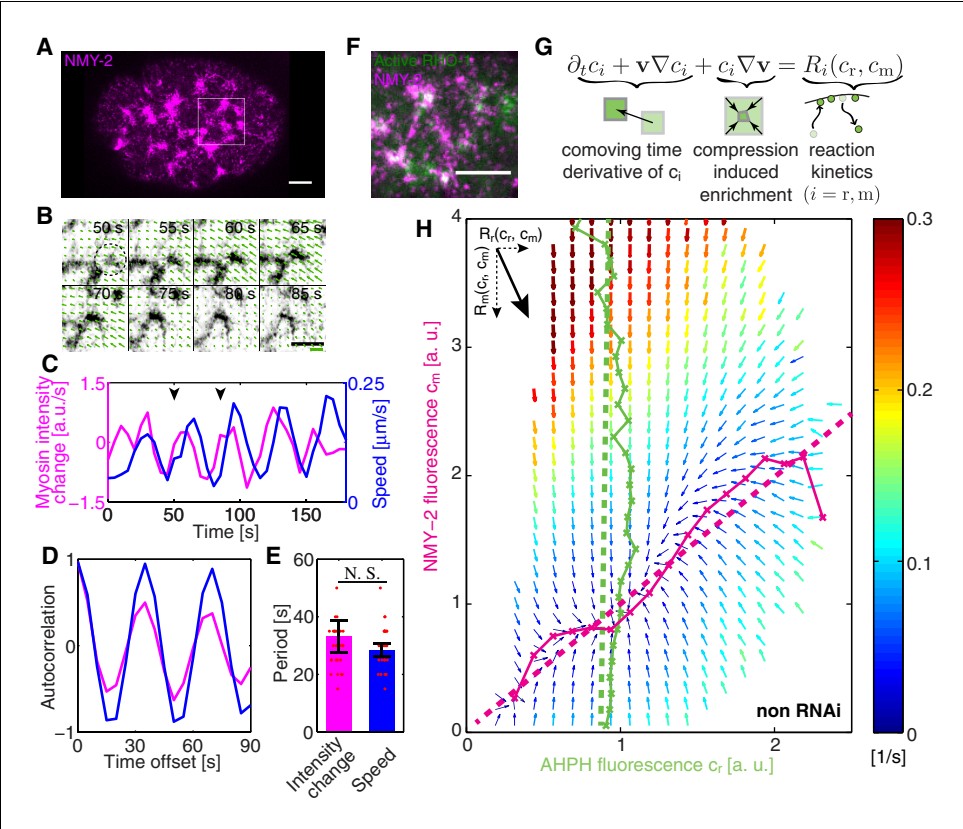

**Figure 1.** COMBI of active RhoA and NMY-2. (**A**) A representative image of NMY-2::GFP showing the NMY-2 foci pattern (magenta) in the *C. elegans* zygote. Anterior is to the left throughout, white box denotes region shown in B. (**B**) Myosin focus assembly and disassembly time-course from **A** in inverted contrast; dashed circle indicates a myosin focus. Arrows denote the velocity field determined by PIV; thick green line: velocity scale bar 0.4 $\mu m/s$. (**C**) The temporal dynamics of NMY-2 fluorescence intensity time-rate change (magenta) and cortical flow speed (blue, obtained by PIV) for the region in (**B**), arrowheads indicate the time interval shown in (**B**). (**D**) Normalized autocorrelation of NMY-2 intensity change and flow speed timecourses in (**C**) and (**E**) respective oscillation periods. (**F**) NMY-2::RFP (magenta) and AHPH::GFP (green), a probe for active RhoA, co-localize at myosin foci. (**G**) COMBI analysis schematic. (**H**) Effective reaction terms of NMY-2 and active RhoA in the phase plane of normalized NMY-2 and active RhoA concentrations (N = 25 embryos). Arrows represent concentration changes, colors indicate the magnitude of change. Thin solid magenta (NMY-2) and green (RhoA) lines, numerically determined nullclines. Thick dashed lines, linearized nullclines (see Appendix). Scale bars; 5 $\mu m$.

The following figure supplements are available for figure 1:

**Figure supplement 1.** Co-localization of active RhoA and myosin.

**Figure supplement 2.** Trajectories in the phase plane of AHPH and NMY-2 concentrations.

**Figure supplement 3.** COMBI of active RhoA and actin.

movement of interrogation areas between two sequential timelapse images (Materials and methods). The spatial resolution of the velocity field is determined by the spacing of the interrogation areas which we choose as $1.26\ \mu m$. This is sufficiently smaller than the correlation length of cortical flow (the hydrodynamic length is ~14 μm [*Mayer et al., 2010*; *Saha et al., 2016*]), hence, COMBI can provide information on actomyosin homeostasis by determining the average reaction kinetics on a timescale of seconds and a length scale of microns.

We visualize the reaction terms determined by COMBI in a vector field that illustrates the average evolution of concentrations of both species (*Figure 1H*). This reveals interesting features, for

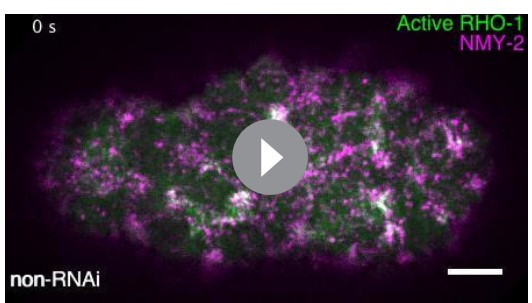

**Video 1.** Active RhoA and myosin co-localization in pulsatile foci. Time lapse movie shows the cortical plane of embryo that expresses both AHPH::GFP (green) and NMY-2::tagRFP-T (magenta) in non-RNAi condition. Scale bar, 5 $\mu m$.

example a trajectory that starts with high active RhoA and low myosin levels will overshoot in its level of myosin prior to approaching the single stable fixed point (thick black line in *Figure 1—figure supplement 2A*). Key aspects of these dynamics can be captured by a nullcline analysis (*Izhikevich, 2007*). Reaction terms are zero on a nullcline ($R_r = 0$ for active RhoA, *Figure 1H*, solid green line; and $R_m = 0$ for myosin, *Figure 1H*, solid magenta line), which describes the concentration that a species would achieve when the concentration of the other species is fixed. This reveals that active RhoA recruits myosin (*Figure 1H*, solid magenta line) (*Motegi and Sugimoto, 2006*; *Schonegg and Hyman, 2006*) while RhoA activation kinetics is essentially independent of myosin levels (*Figure 1H*, solid green line). The full kinetic landscape can be linearized over its entire range (*Figure 1H*, dashed lines; *Figure 1—figure supplement 2B–D*), capturing global aspects of RhoA-based myosin recruitment. To conclude, COMBI can provide insight into the cortical kinetics over a broad range of myosin and active RhoA concentrations.

Given our kinetic analysis, we next sought to test if the full mechanochemically coupled system is indeed unstable. We describe the actomyosin cortex as a thin film of an active gel (*Simha and Ramaswamy, 2002*; *Kruse et al., 2004*; *Ahmadi et al., 2006*; *Salbreux et al., 2009*), with active tension generation by myosin under control of RhoA (*Figure 2A*; see Appendix for details). We measured the relevant material parameters of the gel *in vivo* directly from laser ablation experiments (hydrodynamic length: $\lambda = 14.3\ \mu m$, and a conversion factor from NMY-2 intensity to active tension, $\zeta' = 24.9\ \mu m^2/s$)(*Saha et al., 2016*). This allowed us to perform a linear stability analysis of the homogeneous state for the full model of the mechanochemical patterning system, with the above determined and linearized reaction kinetics between active RhoA and NMY-2 (*Figure 1H*). *Figure 2B* shows the corresponding stability diagram as a function of both the hydrodynamic length of the cortex $\lambda$ and the active tension measure $\bar{\sigma}$ (see Appendix for detail). Notably, the homogeneous state, in which all concentrations are constant in space, always becomes unstable above a critical value of the active tension. Furthermore, we find that the parameter values of the *C. elegans* cortex are such that the system is close to the transition line between stable and unstable, but placed within the unstable regime (*Figure 2B*). Hence, our analysis is consistent with the actomyosin cortex in *C. elegans* being unstable and poised to form a spatial pattern.

Our theory predicts that the contractile instability depends on the strength of positive feedback, and thus the amount of recruitment of myosin by active RhoA (*Figure 2C,D*). Hence, we asked if suppression of RhoA mediated recruitment of myosin in the *C. elegans* zygote prevents the instability and results in a homogeneous NMY-2 distribution. LET-502 is the Rho-associated protein kinase that phosphorylates the regulatory myosin light chain, MLC-4, to activate NMY-2 (*Piekny and Mains, 2002*). Hence, reducing the concentration of LET-502 by RNAi should suppress RhoA mediated recruitment of myosin to the cortex. Indeed, COMBI analysis of *let-502* RNAi embryos (30 hr) revealed that RhoA mediated recruitment of NMY-2 to the cortex is reduced, since the myosin nullcline displays a significantly decreased slope as compared to the non-RNAi condition (*Figure 2E,F*; see *Table 1*). Using the non-RNAi values of $\lambda$ and $\zeta'$ and the linearized reaction kinetics between active RhoA and NMY-2 measured by COMBI in *let-502* RNAi (dark dashed lines in *Figure 2F*), we find that the cortex is predicted to be stable because all eigenvalues are negative (*Figure 2G*, compare to D; see Appendix for detail). Consistent with this prediction, we observed that *let-502* RNAi embryos display a homogeneous NMY-2 distribution without pulsatory myosin foci (*Figure 2H*, compare to *Figure 1A*; *Video 2*). We conclude that, consistent with COMBI and theory, the actomyosin cortex can be brought into a stable regime by reducing positive feedback via suppressing myosin recruitment by active RhoA.

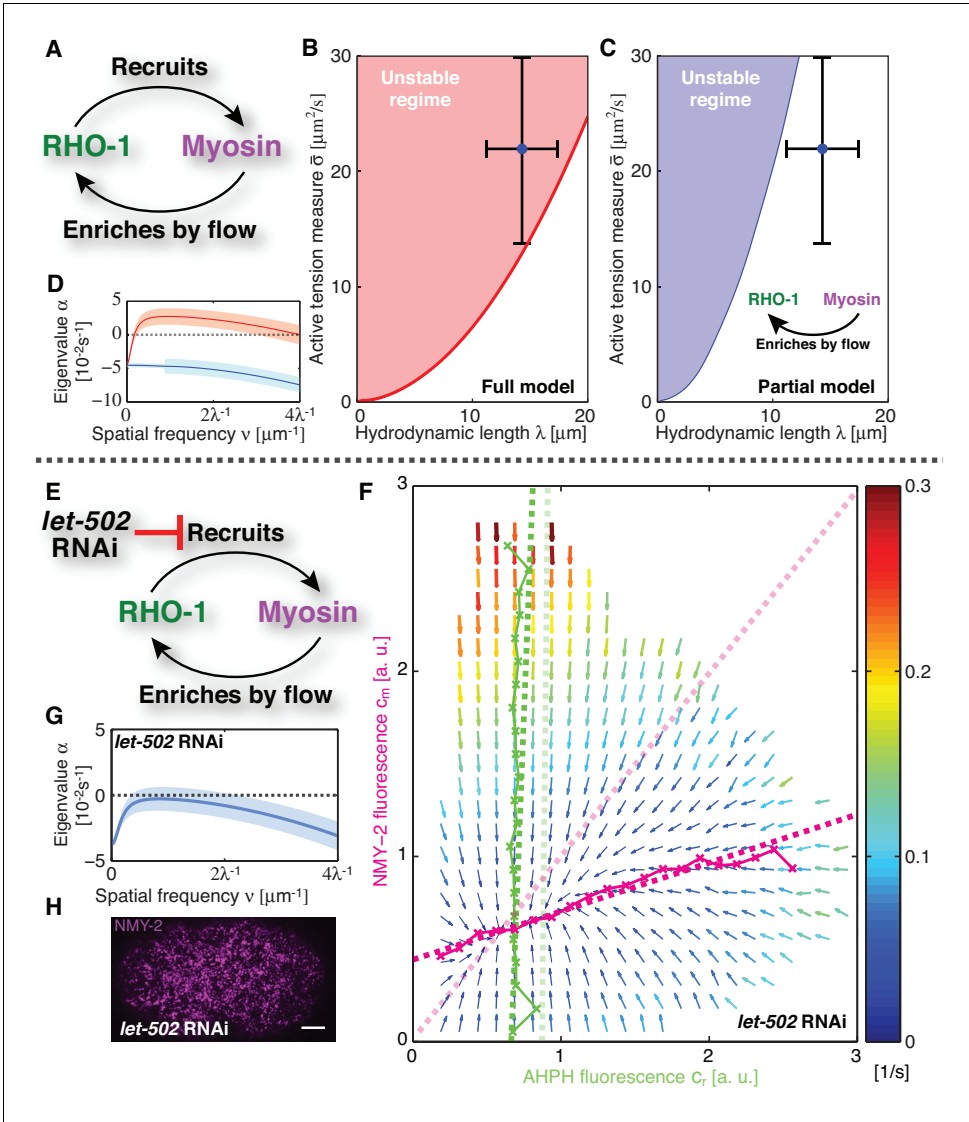

**Figure 2.** Linear stability analysis reveals that the actomyosin cortex in *C. elegans* is unstable. (A) Schematic of the full mechanochemical patterning system. (B) Stability diagram of the homogeneous state in the plane of hydrodynamic length $\lambda$ and active tension measure $\bar{\sigma}$ (see Appendix). The homogeneous state is unstable within the red region. Blue dot represents the parameter values of the non-RNAi *C. elegans* cortex; error bars denote 95% confidence intervals. (C) Stability diagram for a partial model without NMY-2 recruitment by RhoA; inset: corresponding schematic. The homogeneous state is unstable within the blue region. (D) Dispersion relations of the full mechanochemical patterning system with (red) and without (blue) RhoA mediated NMY-2 recruitment. Lighter shared areas represent 95% confidence intervals. (E) *let-502* RNAi suppresses RhoA mediated recruitment of NMY-2. (F) COMBI diagram for *let-502* RNAi (30 hr), N = 12 embryos. Thin solid magenta (NMY-2) and green (RhoA) lines; numerically determined nullclines. Thick solid dashed lines, linearized nullclines. Light dashed lines, linearized nullclines for the non-RNAi condition (***Figure 1H***) for comparison. (G) Dispersion relation for *let-502* RNAi, lighter blue area indicates the 95% confidence interval. (H) NMY-2 distribution under *let-502* RNAi. Scale bar; 5 $\mu m$.

The following figure supplements are available for figure 2:

**Figure supplement 1.** The contractile instability is insensitive to changing the diffusion constants over two orders of magnitude.

**Figure supplement 2.** Stability of the homogeneous state with a linear form of $f(c_{\mathrm{m}}) = c_{\mathrm{m}}$.

**Table 1.** Parameter values.

| Parameters[*,†] | Value |
|---|---|
| **Determined in this study** | |
| **Kinetic parameter for non RNAi** | |
| $k_{on}^{r}$ | $3.96 \pm 0.21 [10^{-2}/s]$ |
| $k_{off}^{r}$ | $4.54 \pm 0.244 [10^{-2}/s]$ |
| $k_{on}^{mr}$ | $0.0576 \pm 0.0934 [10^{-2}/s]$ |
| $k_{on}^{m}$ | $0.126 \pm 0.389 [10^{-2}/s]$ |
| $k_{on}^{rm}$ | $9.94 \pm 0.435 [10^{-2}/s]$ |
| $k_{off}^{m}$ | $10.1 \pm 0.269 [10^{-2}/s]$ |
| **Kinetic parameter for *let*-502 RNAi** | |
| $k_{on}^{r}$ | $2.87 \pm 0.105 [10^{-2}/s]$ |
| $k_{off}^{r}$ | $4.39 \pm 0.0979 [10^{-2}/s]$ |
| $k_{on}^{mr}$ | $0.178 \pm 0.178 [10^{-2}/s]$ |
| $k_{on}^{m}$ | $3.56 \pm 0.165 [10^{-2}/s]$ |
| $k_{on}^{rm}$ | $1.81 \pm 0.0938 [10^{-2}/s]$ |
| $k_{off}^{m}$ | $7.49 \pm 0.106 [10^{-2}/s]$ |
| $D_r, D_m$ | $0.01 [\mu m^2/s]$ |
| **Determined in Saha et al.,** | |
| $\lambda$ | $14.3 \pm 2.94 [\mu m]$ |
| $\zeta/\gamma$ | $24.8 \pm 8.62 [\mu m^2/s]$ |
| **Parameter values for complex Swift-Hohenberg equation** | |
| $a$ | 0.25 |
| $b$ | 0.0000490 |
| $d_1$ | $1.00 + 0.2i$ |
| $d_2$ | $0.0297 + 0.00400i$ |
| $f_0$ | 0.4 |
| $f_1$ | 0.00247 |
| $q_0$ | 10.1 |

*Parameter values are shown with 95 % confidence intervals.

†Active RhoA and NMY-2 densities are normalized by their average concentrations, and reported in dimensionless units of fluorescence intensities per unit area of $1[\text{pixel}]^2$, corresponding to $0.0110 \sim \mu m^2$.

We next asked if the patterns that are formed in the unstable regime in our theory correspond to the pattern of myosin foci observed in the embryo. Earlier work that considers a cortical gel with a diffusible activator of myosin (***Bois et al., 2011***; ***Kumar et al., 2014***) suggests that a contractile instability results in a myosin foci pattern with a spacing that is determined by the hydrodynamic length $\lambda$. Indeed, a numerical solution of the full mechanochemical patterning system (***Figure 2A***) reveals the formation of a few dynamic contracting regions which travel and are spaced about $2\lambda$ apart (***Figure 3—figure supplement 1***, ***Video 3***). These traveling peaks have rapid flows converging upon them (peak flow speed: $0.7 \ \mu m/s$), and they persist and are not pulsatile. This pattern is different from the myosin foci pattern observed in the *C. elegans* zygote, which is pulsatile and exhibits a shorter spacing between foci ($\sim 5 \ \mu m$, compare to $\lambda = 14.3 \ \mu m$; see ***Figure 3—figure supplement 1***) (***Munro et al., 2004***; ***Mayer et al., 2010***). This suggests that our model is missing an essential feature, which is responsible for determining the myosin pattern beyond the contractile instability.

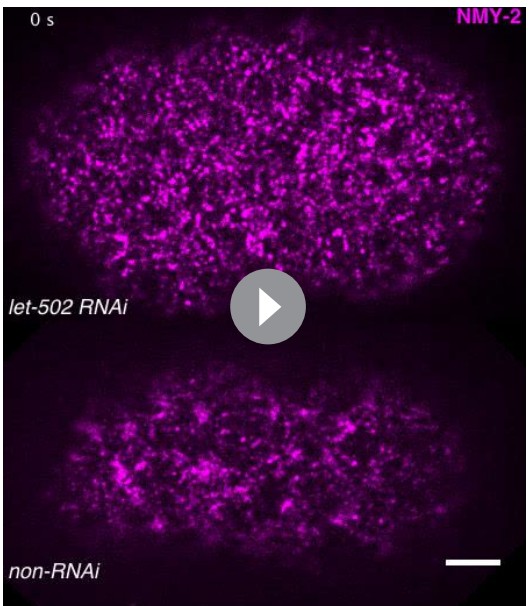

**Video 2.** Homogeneous myosin distribution in suppressed RhoA mediated myosin recruitment. Time lapse movies show the cortical planes of the embryo that expresses NMY-2::tagRFP-T in *let-502* RNAi embryo (upper) and in non-RNAi embryo (lower). Scale bar, 5 $\mu m$.

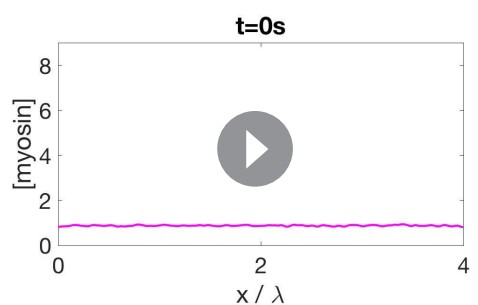

**Video 3.** Traveling peaks of myosin in the mechanochemical patterning system. Time evolution of the myosin pattern, obtained by the numerical integration of the mechanochemical patterning system without an active RhoA pacemaking oscillator.

To identify the element missing in our model, we note that our theory predicts that reducing myosin recruitment by *let-502* RNAi should cause both myosin and RhoA to be homogeneous and non-pulsatile (all eigenvalues are negative, see *Figure 2G*). However, imaging active RhoA under *let-502* RNAi revealed that while the pulsatile myosin pattern is lost, the pulsatile active RhoA pattern is still present (*Figure 3A,C*; *Video 4*). Similarly, we find that 16 hr of RNAi of *nmy-2* led to an almost complete loss of cortical myosin with, however, active RhoA still forming pulsatile foci (*Figure 3B,D*; *Video 5*). We conclude that, in contrast to the scenario in *Drosophila* germband extension (*Munjal et al., 2015*), active RhoA in *C. elegans* exhibits pulsatile foci dynamics independently of NMY-2 function. Importantly, both the characteristic spacing of the myosin-independent active RhoA pattern and its characteristic time-scale were similar between *nmy-2* RNAi, *let-502* RNAi, and the non-RNAi condition (*Figure 3E,F*). Given that active RhoA in the wild-type acts to recruit myosin (*Figure 1H*), this raises the possibility that the myosin-independent dynamic active RhoA pattern is responsible for setting the myosin spatiotemporal pattern beyond the contractile instability. We conclude that the dynamic active RhoA pattern is generated in a manner that is independent of the myosin foci pattern, possibly through an independent RhoA spatiotemporal oscillator.

Oscillatory activities of Rho GTPases have previously been observed (*Hwang et al., 2005*; *Miller and Bement, 2009*; *Das et al., 2012*; *Antoine-Bertrand et al., 2016*). We next asked if this spatiotemporal oscillator requires *ect-2*, a RhoGEF, responsible in the early morphogenesis (*Motegi and Sugimoto, 2006*; *Schonegg and Hyman, 2006*). Indeed, RNAi of *ect-2* leads to a complete absence of RhoA pulsation (*Video 6*). Furthermore, it is interesting to speculate if the myosin-independent active RhoA oscillator that we identify here is related to the RhoA/actin-based excitable oscillatory system reported previously (*Bement et al., 2015*; *Westendorf et al., 2013*). To test if the underlying mechanism to generate myosin-independent active RhoA oscillator is shared between *C. elegans* single-cell embryo and Xenopus embryo, we used COMBI to investigate the effective kinetic regulation between active RhoA and actin. We used LifeAct tagRFP-T as a probe for filamentous actin in the cortex (*Riedl et al., 2008*; *Reymann et al., 2016*). We determined the kinetic diagram in the active RhoA and actin concentration phase plane, to quantitatively evaluate the rate constants in the effective kinetic equations (*Figure 1—figure supplement 3A*). We find that the active RhoA nullcline is nearly vertical and inconsistent with actin behaving as a

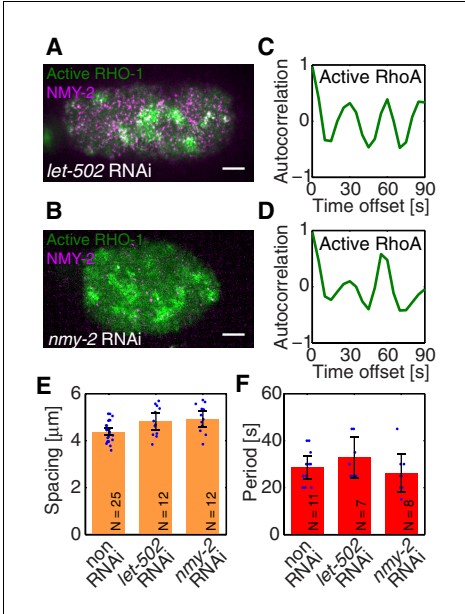

**Figure 3.** Active RhoA exhibits pulsatory dynamics under conditions of reduced myosin activity. (A,B) AHPH::GFP (green) and NMY-2::RFP (magenta) in (A) a representative *let-502* RNAi and (B) a representative *nmy-2* RNAi embryo. (C,D) Normalized AHPH::GFP intensity change autocorrelation (C) for (A) and (D) for B, obtained within the posterior. (E,F) Characteristic (E) spacing of AHPH patterns and (F) period of AHPH intensity change in non-RNAi, *nmy-2* RNAi, and *let-502* RNAi embryos. Scale bars, 5 $\mu m$.

The following figure supplements are available for figure 3:

**Figure supplement 1.** Myosin forms traveling peaks that are spaced approximately $2\lambda$ apart.

**Figure supplement 2.** Characteristic spacing of AHPH foci.

negative regulator of RhoA. Note that this does not exclude the general possibility of negative feedback between actin and RhoA *(Robin et al., 2016)*, but suggests that the *C. elegans* cortex is normally operating in a regime where no such negative feedback is accessed. While the detailed mechanism as well as the kinetic interactions that underlie RhoA pulsation in *C. elegans* remain to be determined, the RhoGEF *ect-2* is involved and the system appears to undergo spatiotemporal oscillations in the absence of negative feedback between actin and RhoA.

We next sought to test in our theory if it is possible that an active RhoA spatiotemporal oscillator sets the myosin pattern beyond the contractile instability (*Figure 4A*, left). To this end, we described the dynamical behavior of an active RhoA pacemaker by use of a generic model of spatiotemporal oscillating patterns, the complex Swift-Hohenberg Equation (*Figure 4—figure supplement 1A*) (*Sakaguchi, 1997*). Importantly, coupling in our model this generic spatiotemporal oscillator ($30 \sim s$ characteristic timescale, $5~\mu m$ characteristic length scale, *Figure 3E,F*; see Appendix for detail) to the full mechanochemical patterning system does not destroy the active RhoA spatiotemporal oscillator pattern. Instead our model predicts that the myosin pattern (which in the absence of the generic oscillator formed a single traveling peak, see *Figure 3—figure supplement 1*) now follows that of the active RhoA spatiotemporal oscillator (*Figure 4B* left). Hence, the active RhoA oscillator can determine the myosin pattern in the unstable regime (*Video 7*). As a consequence, controlling the myosin pattern also results in reduced cortical flow speeds (peak flow speed: 0.17 $\mu m/s$) as compared to the case where the RhoA oscillator is absent (0.7 $\mu m/s$, see above). However, we find that the ability of the RhoA oscillator to control the myosin pattern critically depends on the level of mechanochemical feedback. We demonstrate this by reducing the hydrodynamic length in our model, which increases overall flow speeds and advection, and thereby increases the mechanochemical feedback strength. We find in our model that this change destroys the pattern of the active RhoA spatiotemporal oscillator. Both the myosin and active RhoA pattern no longer form a regular spatiotemporal oscillation (*Figure 4—figure supplement 1*, $\lambda$ is reduced by 5 $\mu m$ to 9 $\mu m$). Instead, the system displays a dynamical state that is characterized by an irregular spatiotemporal pattern of dynamic contracting regions that move rapidly (*Figure 4B* right; *Figure 4—figure supplement 1*; *Video 8*). In this state, the pattern of active RhoA now depends on myosin and flows and is essentially under control of the contractile instability. Finally, flow speeds are again increased and comparable to the case when the RhoA oscillator is absent (peak flow speed: 0.94 $\mu m/s$). In conclusion, theory indicates that the active RhoA oscillator can act as a pacemaker for the system, to control the contractile instability and to prevent the formation of large and irregularly moving contracting regions of myosin.

We next sought to seek experimental evidence that the myosin pattern in the *C. elegans* zygote is under control of the RhoA pacemaker. To this end, we tested if increasing the level of mechanical

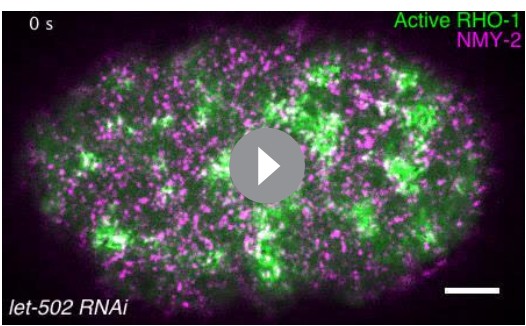

**Video 4.** Pulsatile dynamics of the active RhoA exhibits pulsatile dynamics in homogenous myosin cortex. Time lapse movie shows the cortical plane of the embryo that expresses both AHPH::GFP (green) and NMY-2:: tagRFP-T (magenta) in *let-502* RNAi embryo. Scale bar, 5 *μm*.

feedback in *C. elegans* destroys the pattern of the active RhoA spatiotemporal oscillator as predicted from theory. For this we recorded space-time patterns of myosin in a midplane section under conditions of *spd-5* RNAi. SPD-5 is a centriole constituent that is essential for centriole maturation (*Hamill et al., 2002*), and its RNAi leads to a delay of polarizing flows which gives us more time for an analysis of the pulsatory dynamics. To increase mechanochemical feedback we recorded space-time patterns of myosin under RNAi of the actin nucleator *pfn-1* (*Severson et al., 2002*) for which the cortex is weakened and flow speeds are increased by a factor of three to five (*Figure 4F*) and for which the hydrodynamic length is decreased by 5 *μm* to 9 *μm* (*Severson et al., 2002*; Naganathan, unpublished). For *spd-5* RNAi, we observed the 'normal' pulsating pattern of myosin foci (*Figure 4C* left; *Video 9* and *Video 10*).

In contrast, for *pfn-1* RNAi we observed large contracting regions of myosin that rapidly move in an irregular fashion (*Figure 4C* right, D right; *Videos 9* and *11*). Importantly, in *pfn-1* RNAi active RhoA assembles in large and irregularly moving foci structures (*Figure 4E* right, compare to *Figure 4E* left and *Figure 3A,C*; *Video 11*). This suggests that the normal pattern of the active RhoA spatiotemporal oscillator is destroyed, and the distribution of active RhoA is now governed by the irregular dynamics of myosin. Note that *pfn-1* RNAi does not destroy the general ability of RhoA to generate a pulsating pacemaker pattern, as revealed by double RNAi of *pfn-1* and *nmy-2* (*Video 12*). We conclude that reducing the hydrodynamic length increases mechanochemical feedback and advection. This causes the RhoA pacemaker to lose the ability to control the contractile instability. Consistent with the predictions of our theory, this destroys the RhoA pacemaker pattern and causes the system to undergo a transition to irregular behavior with large and rapidly moving contracting regions (*Figure 4C–E*). Interestingly, the uncontrolled gel is no longer capable to drive coherent flows of the cortex over large distances (*Figure 4G*), and the embryo fails to polarize (*Severson et al., 2002*). Taken together, our quantitative analysis is consistent with the interpretation that the spatiotemporal RhoA oscillator acts as a pacemaker in *C. elegans*, controlling the contractile instability of the actomyosin cortex.

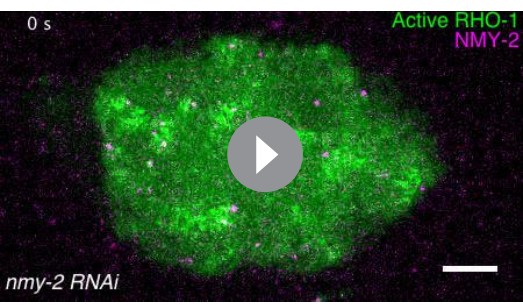

**Video 5.** Pulsatile dynamics of the active RhoA exhibits pulsatile dynamics independently of myosin function. Time lapse movie shows the cortical plane of the embryo that expresses both AHPH::GFP (green) and NMY-2::tagRFP-T (magenta) in *nmy-2* RNAi embryo. Scale bar, 5 *μm*.

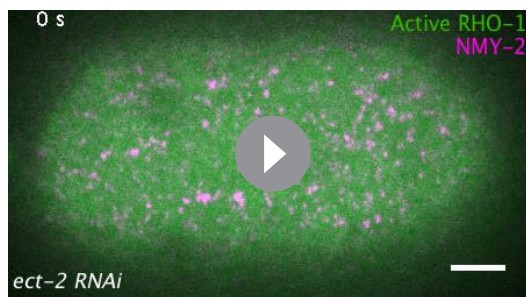

**Video 6.** The absence of the active RhoA pacemaking oscillator in *ect-2* RNAi embryo. Time lapse movies show the cortical planes of the embryos that expresses AHPH::GFP (green) in *nmy-2* RNAi embryo (upper), and in *ect-2* RNAi embryo (lower). Scale bar, 5 *μm*.

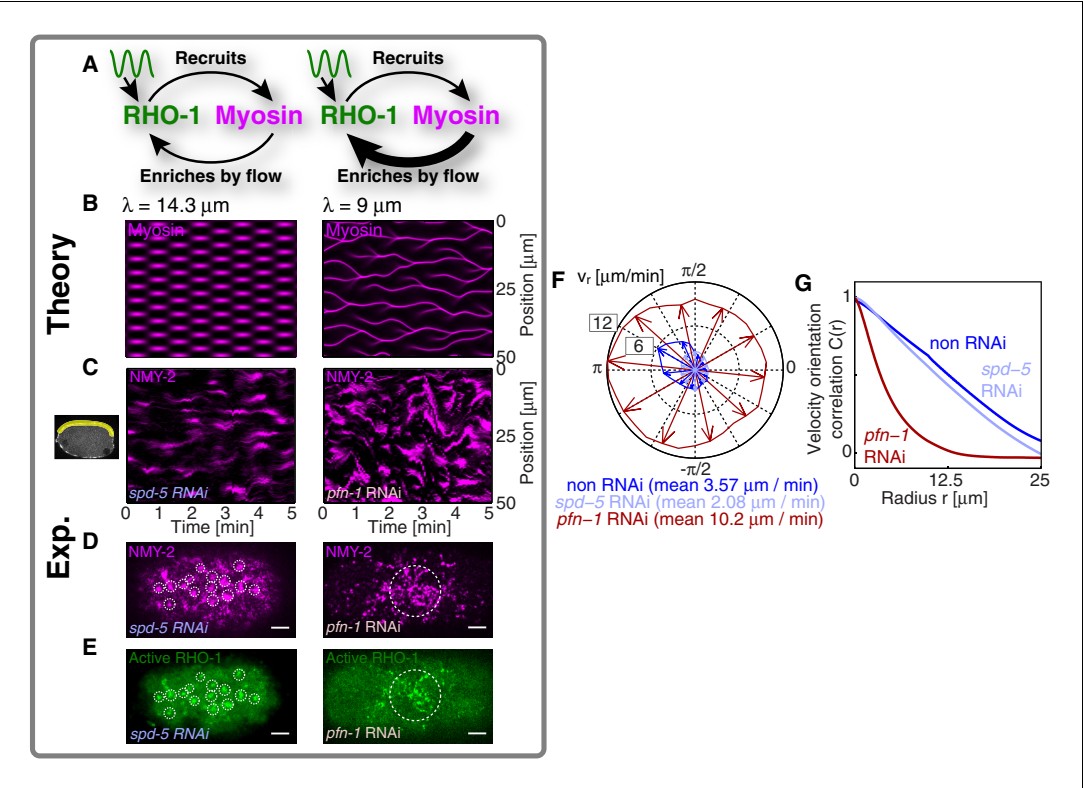

**Figure 4.** A RhoA pacemaking oscillator controls the contractile instability. (**A**) Schematic of a mechanochemical patterning system under control of a RhoA pacemaker, with (left) normal conditions and (right) with increased mechanochemical feedback and with faster flows. (**B**) Numerically obtained space time plots of the myosin distribution, for normal conditions (λ = 14.3 μm; left) and for a weakened cortex with increased mechanochemical feedback (λ = 9 μm; right); see Appendix. (**C**) Kymographs of NMY-2 intensity under normal conditions (*spd-5* RNAi; left) and under conditions of a weakened cortex (*pfn-1* RNAi; right) obtained in mid-plane images and from the yellow region illustrated in the inset image on the right. (**D,E**) Representative cortical plane images of (**D**) NMY-2::tagRFP-T and (**E**) RhoA::GFP, dotted circles indicate foci. (**F**) Average cortical flow speed as a function of direction under conditions of a normal cortex (dark blue: non-RNAi; light blue: *spd-5* RNAi) as well as for a weakened cortex (red: *pfn-1* RNAi). (**G**) Radially averaged velocity orientation correlation function (Materials and methods) for the same three conditions, note that the *pfn-1* RNAi embryo cannot drive coherent flow over large distances. Scale bars, 5 μm.

The following figure supplement is available for figure 4:

**Figure supplement 1.** A RhoA pacemaking oscillator can control the myosin pattern in the model.

We have here investigated the mechanisms of pattern formation in an active system that combines the contractile force generation and flow with regulation and advection. For this, we introduced the COMBI method to directly infer reaction kinetics without relying for example on photobleaching (*Sprague et al., 2004*). We determined the effective reaction kinetics of myosin and active RhoA in the actomyosin cortex with COMBI. This allowed us to build a quantitative model of mechanochemical patterning in the actomyosin layer. By use of linear stability analysis, we found that the actomyosin cortex is unstable and spontaneously forms a self-organized pattern. We speculate that during embryogenesis cells need high cortical contractility to drive morphological changes. This can lead them near or beyond contractile instabilities, leading to dynamics characterized by strong fluctuations and irregular behavior, possibly exhibiting active turbulence (*Giomi, 2015*). We suggest that such instabilities are inevitable in dynamic systems that are highly contractile. We discovered a spatiotemporal RhoA oscillator that determines the myosin pattern even beyond the contractile instability, thereby controlling the contractile instability. The independent biochemical RhoA

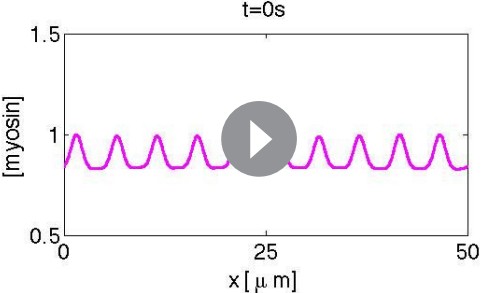

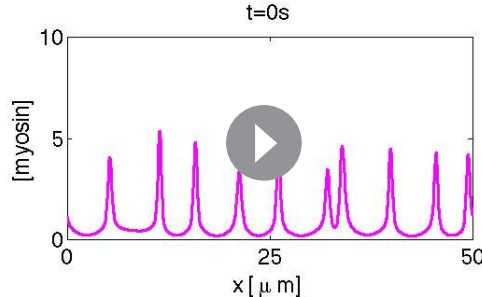

**Video 7.** Active RhoA pacemaker can determine the myosin pattern in the contractile instability regime. Time evolution of the myosin pattern, obtained by the numerical integration of the mechanochemical patterning system, coupled with the active RhoA pacemaking oscillator.

**Video 8.** Rapid and irregularly moving myosin pattern in the cortex with the reduced hydrodynamic length. Time evolution of the myosin pattern, obtained by the numerical integration of the mechanochemical patterning system with the reduced hydrodynamic length by 5 $\mu m$ to 9 $\mu m$, and coupled with the active RhoA pacemaking oscillator.

oscillator endows the cell with the ability to use an intrinsically unstable active contractile medium for driving morphogenetic processes such as polarization. To conclude, our work paves the way for understanding pattern formation in active biological materials that utilize potentially unstable contractile processes.

# Materials and methods

### Worm strains, maintenance, and sample preparation

The following transgenic lines were used in this study. SWG003: nmy-2(cp8[nmy-2::GFP + unc-119(+)]) I; unc-119(ed3) III; gesIs002[Ppie-1::Lifeact::tagRFP-T::pie-1UTR + unc-119(+)], for imaging of GFP labelled NMY-2 (the images shown in *Figure 1A and B* and in *Figure 4C*). SWG012: nmy-2(ges6[nmy-2::tagRFP-T + unc-119(+)]) I; xsSi5[cb-UNC-119 (+) GFP:: ANI-1 (AH+PH)] II; unc-119(ed3) III, for imaging of tagRFP-T labelled NMY-2 and GFP labelled AHPH for a probe of active RhoA in the cortex (the images shown in *Figure 1F*, in *Figure 2H*, and in *Figure 3A and B*).

Worm strains were maintained at 20°C, and shifted to 24°C for 24 hr before imaging. Embryos were dissected in M9 buffer and mounted onto agar pads (2% agarose in water) to squish the embryos gently. All experiments were performed at 23–24°C. RNA interference experiments were performed by feeding as described in *Naganathan et al. (2014)*. Feeding times for RNAi experiments were 16–18 hr for *nmy-2*, 23–25 hr for *spd-5*, 19–21 hr for *pfn-1*,

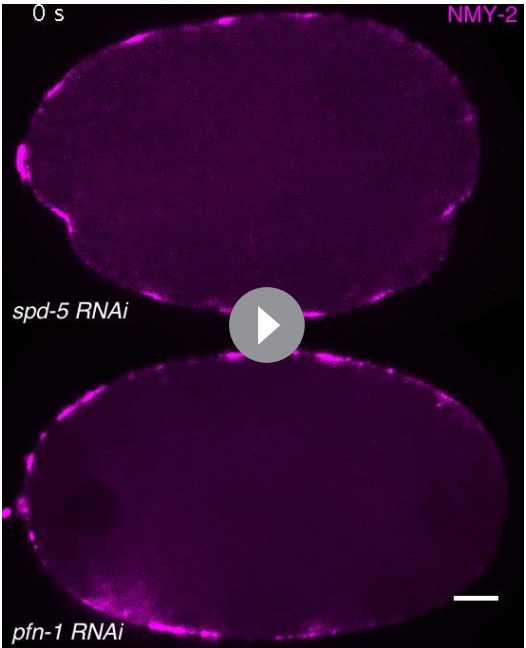

**Video 9.** Rapid and irregular movement of myosin foci for a *pfn-1* RNAi embryo. Time lapse movies show the midplane sections of the embryos that express NMY-2::GFP (magenta) in *spd-5* RNAi embryo (upper) and *pfn-1* RNAi embryo (lower). Scale bar, 5 $\mu m$.

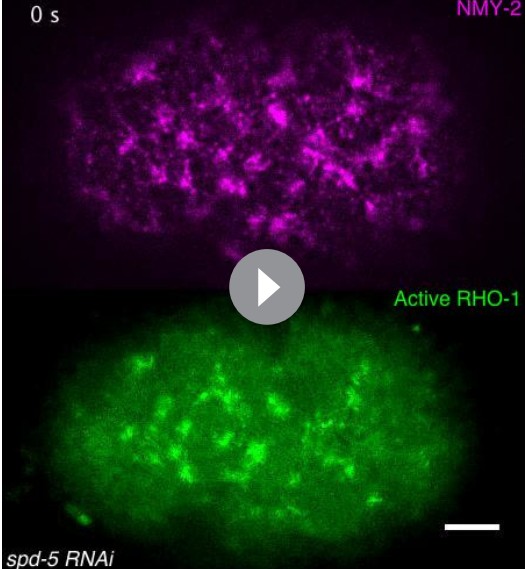

**Video 10.** Pulsatile dynamics of the active RhoA and the myosin in *spd-5* RNAi embryo. Time lapse movies show the cortical planes of the embryo that expresses both AHPH::GFP (green) and NMY-2::tagRFP-T (magenta) in *spd-5* RNAi embryo. Scale bar, 5 $\mu m$.

**Video 11.** Irregular dynamics of the active RhoA and the myosin in *pfn-1* RNAi embryo. Time lapse movies show the cortical planes of the embryo that expresses both AHPH::GFP (green) and NMY-2::tagRFP-T (magenta) in *pfn-1* RNAi embryo. Scale bar, 5 $\mu m$.

19–21 hr for *pfn-1;nmy-2* double knockdown and 29–31 hr for *let-502*. Feeding clones were obtained from the Hyman lab (MPI-CBG, Dresden, Germany).

## Imaging

One-cell stage embryos were observed under the inverted fluorescence microscope (Axio Observer Z1, Zeiss) using a Zeiss C-Apochromat 63× water immersion lens, equipped with a spinning disc confocal unit (Yokogawa, CSU-X1) and AOTF laser combiner (Andor, ALC). Fluorescence images were acquired by a sCMOS camera (Hamamatsu, ORCA flash 4.0) at 5 s time intervals for non-RNAi, *let-502* RNAi, *nmy-2* RNAi, and *pfn-1;nmy-2* RNAi embryos. For *pfn-1* RNAi embryos, images were taken every 3 s. Pixel size was 0.105 × 0.105 $\mu m^2$, all devices were controlled through $\mu-$manager (*Edelstein et al., 2014*). Fluorescence images of GFP and tagRFT-T labeled proteins in the embryos were excited by 488 and 561 nm lasers, respectively.

## Image analysis

Prior to COMBI analysis, images were filtered using the nonlocal means method (*Buades et al., 2005*), reducing spatially uncorrelated noise while preserving finer structures. Filtering was performed by averaging fluorescence intensities on the basis of the similarity between the fluorescence intensity profile in the interrogation area and the intensity profile in the neighboring region, i.e., a searching window. We set the size of the interrogation area and the searching window, to be 5 × 5 pixels and 25 × 25 pixels, corresponding to 0.525 × 0.525 $\mu m^2$ and 2.625 × 2.625 $\mu m^2$, respectively. A filtering parameter, $h$, was set to be 0.1 $s$ for NMY-2 images, 0.3 $s$ for AHPH images, where $s$ denotes the standard deviation of the fluorescence intensity in each image. We performed the filtering using a freely available code from MATLAB central (Fast Non-Local Means 1D, 2D Color and 3D by Kroon). Note that it is important to remove spatially uncorrelated noise prior to the computation of the spatial derivatives, since differential value is affected by spatially uncorrelated noises.

To perform COMBI, we first determined the cortical flow velocity, $\mathbf{v}(x, y)$, by Particle Image Velocimetry (PIV) using a freely available PIV algorithm, PIVlab 1.32 (available from http://pivlab.blogspot.

de/). PIV was performed on NMY-2 images by setting the interrogation area as 24 pixels with a step of 12 pixels. Velocity vectors were then interpolated to single pixel resolution for determining $R_r$ and $R_m$. $\partial_t C_i$, $C_i \nabla \mathbf{v}$ and $\mathbf{v} \nabla C_i$ for both the background subtracted, active RhoA and myosin intensities (denoted by $C_r$ and $C_m$, respectively). Intensity background levels were obtained by averaging the intensity in the region outside the embryo. $R_r$ and $R_m$ were then determined for each pixel throughout the cortical plane, by the use of the mass balance equations given in *Figure 1G*. We obtained a kinetic diagram (*Figure 1H*) by averaging $R_r$ and $R_m$ in $10 \times 10\ \mu m^2$ boxes located in the anterior region. We determined average values for each embryo by averaging over the first 36 frames after the start of polarizing flow. For the non-RNAi case (*Figure 1H*), we report the average kinetic diagram from $N = 25$ embryos, for *let-502* RNAi (*Figure 2F*) we averaged over $N = 12$ embryos. Note that the active RhoA and NMY-2 concentrations were normalized by the respective mean intensities of active RhoA and NMY-2 under non-RNAi conditions.

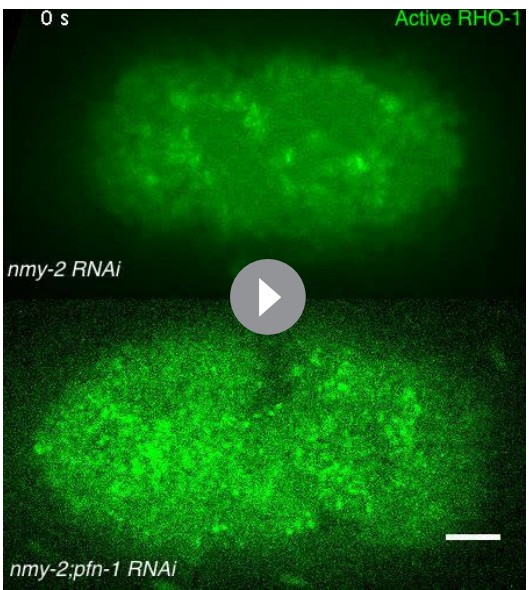

**Video 12.** Pulsatile dynamics of the active RhoA in *pfn-1;nmy-2* RNAi embryo. Time lapse movies show the cortical planes of the embryos that expresses AHPH::GFP (green) in *nmy-2* RNAi embryo (upper), and in *pfn-1;nmy-2* RNAi embryo (lower). Scale bar, 5 $\mu m$.

## Correlation analysis

To characterize the myosin intensity change, $\Delta I(t)$ and the cortical flow speed $v_r(t)$ in a box of size $10 \times 10\ \mu m^2$ in the posterior region, we determined the spatial average over the box. We then computed the autocorrelation function

$$C_t(\tau) = \frac{\left\langle \left[ f(t)^2 - \bar{f} \right] \left[ f(t+\tau)^2 - \bar{f} \right] \right\rangle_t}{\sigma_f^2}, \tag{1}$$

where $f(t) = \Delta I(t)$ or $v_r(t)$, $\sigma_f$ denote the standard deviation of $f(t)$, and $\bar{f}$ denotes the mean of $f(t)$, averaged over time $t$, where $\langle \rangle_t$ represents an average over time. $C_t$ is mean-subtracted and normalized by the variance of $f(t)$. The period of oscillation was determined by the peak position in the autocorrelation function. For a precise detection of oscillatory behavior, we removed from our analysis embryos in which the second peak in the autocorrelation function of the time course was undetectable (12 out of 25 cases for non-RNAi embryos, 5 out of 12 cases in *let-502* embryos, 4 out of 12 embryos for *nmy-2* RNAi embryos, respectively).

We obtained the characteristic length of the spatial pattern of myosin and RhoA by detecting the location of the first peak in the radial spatial intensity correlation function. The spatial intensity autocorrelation function of intensity, $f(x, y, t)$, was obtained by

$$C_{sp}(\xi, \eta, t) = \frac{\left\langle \left[ f(x,y,t)^2 - \bar{f}(t) \right] \left[ f(x+\xi, y+\eta, t)^2 - \bar{f} \right] \right\rangle_{x,y}}{\sigma_f^2}, \tag{2}$$

where $\bar{f}(t)$ denotes the spatial average of $f(x,y,t)$, see *Figure 3—figure supplement 2C*. This function was radially averaged, and the first peak was detected (*Figure 3—figure supplement 2D*) in each time point. The radii of first peak were then averaged over time in each embryo. For the determination of the spacing in AHPH foci, the contrast of the fluorescence images was enhanced using the Contrast Limited Adaptive Histogram Equalization method using Matlab (Mathworks)(see *Figure 3—figure supplement 2*).

To characterize the spatial coherence of the velocity field, we evaluated the spatial correlation of the normalized velocity vectors, $\mathbf{n}(x,y,t) = \mathbf{v}(\mathbf{x},\mathbf{y},\mathbf{t})/||\mathbf{v}(\mathbf{x},\mathbf{y},\mathbf{t})||$. The spatial correlation function was computed by,

$$C_{\mathrm{ori}}(\xi, \eta) = \langle \mathbf{n}(x,y,t) \cdot \mathbf{n}(x+\xi, y+\eta, t) \rangle_{x,y,t}, \tag{3}$$

where $\cdot$ represents scalar product, and $\langle\rangle_{x,y}$ denotes the spatial average. Note that the coordinate transformation from Cartesian to polar coordinates of the orientation vectors, $\mathbf{n}(x,y,t)$, provides a simpler representation of $C_{\mathrm{ori}}(\xi, \eta)$ as,

$$C_{\mathrm{ori}}(\xi, \eta) = \langle \cos[\theta(x,y,t) - \theta(x+\xi, y+\eta, t)] \rangle,$$

where $\theta(x,y,t)$ denotes the anti-clockwise angle from x-axis of $\mathbf{n}(x,y,t)$. The above expression shows that the $C_{\mathrm{ori}}(\xi, \eta)$ provides spatial correlation of the cosine similarity between $\theta(x,y,t)$ and $\theta(x+\xi, y+\eta, t)$. Therefore, the characteristic length of the decay of $C_{\mathrm{ori}}(\xi, \eta)$ represents the loss of correlation between the directions of velocity vectors, $(\xi, \eta)$ away. Larger characteristic length of the decay demonstrates the large-scale flow of the cortex.

For visualizing purpose, we transformed the coordinate system from $(\xi, \eta)$ to polar $(r, \phi)$ and then determined the average over the angle, $\phi$, to plot $C_{\mathrm{ori}}$ as a function of the radius, $r$, e.g. *Figure 4G*.

## Acknowledgements

We are grateful to Michael Glotzer and Daniel Dickinson for providing *C. elegans* strains. We acknowledge Stefan Eimer for the gift of tagRFP-T gene. We thank G Salbreux, JS Bois, PGross and P Gönczy for fruitful discussions. SWG was supported by the DFG (SPP 1782, GSC 97, GR 3271/2, GR 3271/3, GR 3271/4), the European Research Council (grant No 281903), ITN grants 281903 and 641639 from the EU, the Max-Planck-Society as a Max-Planck-Fellow, and the Human Frontier Science Program (RGP0023/2014).

## Additional information

### Competing interests
FJ: Reviewing editor, *eLife*. The other authors declare that no competing interests exist.

### Funding

| Funder | Grant reference number | Author |
|---|---|---|
| Deutsche Forschungsgemeinschaft | SPP 1782 | Stephan W Grill |
| European Research Council | 281903 | Stephan W Grill |
| Human Frontier Science Program | RGP0023/2014 | Stephan W Grill |
| European Commission | ITN grant - 281903 | Stephan W Grill |
| Max-Planck-Gesellschaft | | Stephan W Grill |
| European Commission | ITN grant - 641639 | Stephan W Grill |
| Deutsche Forschungsgemeinschaft | GSC 97 | Stephan W Grill |
| Deutsche Forschungsgemeinschaft | GR 3271/2 | Stephan W Grill |
| Deutsche Forschungsgemeinschaft | GR 3271/3 | Stephan W Grill |
| Deutsche Forschungsgemeinschaft | GR 3271/4 | Stephan W Grill |

The funders had no role in study design, data collection and interpretation, or the decision to submit the work for publication.

**Author ORCIDs**

Masatoshi Nishikawa, http://orcid.org/0000-0001-6502-7907

Frank Jülicher, http://orcid.org/0000-0003-4731-9185

Stephan W Grill, http://orcid.org/0000-0002-2290-5826

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

## Appendix

### Active gel description of the cortex

We described the dynamics of the cortical layer on the surface of the zygote by use of a full mechanochemically coupled system containing an active gel description for the cortical layer (**Bois et al., 2011**; **Kumar et al., 2014**) together with a reaction-diffusion-advection equation system for active RhoA and NMY-2. For the latter, the concentrations of active RhoA and NMY-2 in one dimension are given by $\rho_{\mathrm{m}}(x,t)$ and $\rho_{\mathrm{r}}(x,t)$, respectively, where $x$ denotes position and $t$ denotes time. The dynamics of the 1-D surface concentration vector $\boldsymbol{\rho}(x,t) = [\rho_{\mathrm{m}}(x,t), \rho_{\mathrm{r}}(x,t)]^t$ is given by

$$\partial_t \rho_i(x,t) + \partial_x j_i(x,t) = R_i(\boldsymbol{\rho}(x,t)) \tag{4}$$

$$j_i(x,t) = v(x,t)\rho_i(x,t) - D_i\partial_x\rho_i(x,t), \tag{5}$$

where $i \in \{m, r\}$, with $m$ denoting NMY-2 and $r$ denoting active RhoA. Here, $v(x,t)$ represents the gel velocity field determined by active cortical mechanics, see below. $R_{\mathrm{m}}(\boldsymbol{\rho}(x,t))$ and $R_{\mathrm{r}}(\boldsymbol{\rho}(x,t))$ denote the respective fluxes of NMY-2 and active RhoAand from the cytosol onto the surface via turnover and recruitment. $D_{\mathrm{m}}$ and $D_{\mathrm{r}}$ denote the respective diffusion coefficients of active RhoA and NMY-2. We estimated the surface diffusion coefficients of both active RhoA and myosin, by analyzing FRAP recovery in ANI-1::GFP and NMY-2::GFP by use of a method considers of both turnover and lateral diffusion (**Goehring et al., 2010**). This revealed in both cases that lateral (surface) diffusion was below the detection limit and undistinguishable from zero. Hence, we expect surface diffusion of myosin and active RhoA to not significantly impact the dynamics of the cortical layer. However, for the purpose of preventing sharp peaks in concentration fields in our numerical simulations, we here set $D_{\mathrm{r}} = D_{\mathrm{m}} = 0.01\ \mu m^2/s$. Note that the general results of our numerical analysis do not change when increasing or decreasing either diffusion constant by a factor of 10. Importantly, the fastest growing mode occurs for wavelengths of the order of $\lambda$ in all cases (**Figure 2—figure supplement 1**).

We describe forces and flows within the cortical layer in the framework of active gel theory (**Bois et al., 2011**; **Kumar et al., 2014**; **Kruse et al., 2005**; **Mayer et al., 2010**). We consider the cortex to be a thin film active viscous fluid in 1D, where NMY-2 generates active tension (**Bois et al., 2011**; **Kumar et al., 2014**; **Mayer et al., 2010**). Hence, total tension

$\sigma$ in the layer is given by a sum of viscous tension arising characterized by a bulk viscosity $\eta$, and active tension (or contractility) $\sigma^a$ that depends on myosin concentration acording to $\sigma^a = \zeta f(\rho_{\mathrm{m}})$, with $f(\rho_{\mathrm{m}})$ and increasing function of the concentration $\rho_{\mathrm{m}}$ and $\zeta$ a coefficient that determines the magnitude of active tension.

The constitutive equation for an active viscous fluid is then given by **Mayer et al. (2010)**

$$\sigma = \eta\partial_x v(x,t) + \sigma^a(\rho_m). \tag{6}$$

We choose for $f(\rho_{\mathrm{m}})$ a saturating Hill-function, $f(\rho_{\mathrm{m}}) = \rho_{\mathrm{m}}(\rho_{\mathrm{m}0} + 1)/(\rho_{\mathrm{m}} + 1)$(**Bois et al., 2011**; **Kumar et al., 2014**), to limit the active stress in our numerical simulations. Here, $\rho_{\mathrm{m}0}$ denotes the stationary concentration of NMY-2. Note that choosing a linear dependence without saturation does not significantly change the stability diagram, see **Figure 2—figure supplement 2**.

The force balance equation for an active viscous fluid in the presence of friction with a coefficient $\gamma$ between the cortex and its surrounding cytoplasm and cell membrane (**Mayer et al., 2010**; **Bois et al., 2011**; **Kumar et al., 2014**) is given by

$$\partial_x \sigma = \gamma v. \tag{7}$$

*Equations 6 and 7* provide an equation of motion

$$\lambda \partial_x^2 v + \frac{\zeta'}{\lambda} \partial_x f(\rho_{\mathrm{m}}) - \frac{1}{\lambda} v = 0, \tag{8}$$

where $\lambda = \sqrt{\eta/\gamma}$ represents the hydrodynamic length of the cortex, which sets a correlation length for the velocity field. We determined the parameters $\lambda$ and $\zeta' = \frac{\zeta}{\gamma}$ that characterize the active viscous fluid with a method that compares the relaxation dynamics of the cortex in response to cortical laser ablation (COLA) between experiment and theoretical predictions ($\lambda = 14.3\ \mu m$, $\zeta' = 24.9\ \mu m^2/s$) (**Saha et al., 2016**). Note that we ignored the polar or nematic order of actin filament in the cortex, which can introduce the tension anisotropy. Based on the recent study by Reymann et al. (**Reymann et al., 2016**), the alignment is mainly due to large-scale cortical flow in one-cell stage *C. elegans* embryo, which has the highest compressible component and nematic order parameter in the mid-zone of posterior.

## Rescaling

We rescale the spatial coordinate with respect to $\lambda$, and rewrite *Equations 5–7* according to

$$\begin{aligned}
\tilde{\sigma} &= \partial_{\tilde{x}} \tilde{v} + \tilde{\sigma}^a \\
\partial_{\tilde{x}} \tilde{\sigma} &= \tilde{v} \\
\partial_t \rho_i + \partial_{\tilde{x}} \tilde{v} \rho_i - \tilde{D}_i \partial_{\tilde{x}}^2 \rho_i &= R_i(\rho_{\mathrm{m}}, \rho_{\mathrm{r}}),
\end{aligned} \tag{9}$$

with the following rescaled quantities

$$\begin{aligned}
\tilde{x} &= x/\lambda \\
\tilde{v} &= \frac{v}{\lambda} \\
\tilde{\sigma} &= \frac{\sigma}{\lambda^2 \gamma} \\
\tilde{\sigma}^a &= \frac{\zeta}{\lambda^2 \gamma} f(\rho_{\mathrm{m}}) \\
\tilde{D}_i &= \frac{D}{\lambda^2}.
\end{aligned}$$

We performed numerical integrations of *Equation 9* with periodic boundary conditions by use of the pseudospectral method (**Boyd, 2001**).

## Linear stability analysis

The homogeneous state with the concentrations $\boldsymbol{\rho}_0 = (\rho_{\mathrm{r}0}, \rho_{\mathrm{m}0})^t$ and a vanishing flow field $v(x,t) = 0$ can be a stationary state of the system. To test the stability of this stationary state, we apply a small perturbation of $\boldsymbol{\delta\rho} = \boldsymbol{\rho}_0 + \boldsymbol{\delta\rho}_0\ e^{2\pi i \nu x + \alpha(\nu)t}$, where $\nu$ and $\alpha$ denote spatial frequency and eigenvalue, respectively. Inserting $\boldsymbol{\delta\rho}$ into *Equation 9* and retaining linear terms only, the linear stability matrix $\mathbf{A}$ with respect to the perturbation $\boldsymbol{\delta\rho}$ becomes

$$\mathbf{A} = 4\pi^2 \nu^2 \begin{bmatrix} -\tilde{D}_{\mathrm{r}} & 0 \\ 0 & -\tilde{D}_{\mathrm{m}} \end{bmatrix} + \frac{4\pi^2 \nu^2 \tilde{\zeta} \partial_{\rho_{\mathrm{m}}} f(\rho_{\mathrm{m}0})}{4\pi^2 \nu^2 + 1} \begin{bmatrix} 0 & \rho_{\mathrm{r}0} \\ 0 & \rho_{\mathrm{m}0} \end{bmatrix} + \boldsymbol{\Omega} \tag{10}$$

$$\boldsymbol{\Omega} = \begin{bmatrix} \partial_{\rho_{\mathrm{r}}} R_{\mathrm{r}}(\boldsymbol{\rho}_0) & \partial_{\rho_{\mathrm{m}}} R_{\mathrm{r}}(\boldsymbol{\rho}_0) \\ \partial_{\rho_{\mathrm{r}}} R_{\mathrm{m}}(\boldsymbol{\rho}_0) & \partial_{\rho_{\mathrm{m}}} R_{\mathrm{m}}(\boldsymbol{\rho}_0) \end{bmatrix}, \tag{11}$$

where $\tilde{\zeta} = \zeta/\lambda^2\gamma$. To determine the linearized matrix $\boldsymbol{\Omega}$ that characterizes biochemical regulation of NMY-2 and active RhoA, we performed a linear regression over the whole range of $R_r$ and $R_m$ values determined by COMBI (**Figure 1H** in main text) using

$$
\begin{aligned}
R_r(c_r, c_{rm}) &= k_{on}^r - k_{off}^r c_r + k_{on}^{mr} c_m \\
R_m(c_r, c_{rm}) &= k_{on}^m + k_{on}^{rm} c_r - k_{off}^m c_m .
\end{aligned}
\tag{12}
$$

Here, $c_r$ and $c_m$ denote the normalized intensities of AHPH and NMY-2, respectively, with $[c_r, c_m] = [\rho_r, \rho_m]$. We get

$$
\boldsymbol{\Omega} = \begin{bmatrix} -k_{off}^r & k_{on}^{mr} \\ k_{on}^{rm} & -k_{off}^m \end{bmatrix}.
\tag{13}
$$

We determined the matrix $\boldsymbol{\Omega}$ directly from our experimental data (**Figure 1H** in main text) by robust regression and using standard deviations for weights, and find

$$
\boldsymbol{\Omega} = \begin{bmatrix} -0.0454 & 0.000576 \\ 0.0994 & -0.101 \end{bmatrix}.
\tag{14}
$$

Please see **Figure 1—figure supplement 2** for a comparison between the linearized and full landscapes. The corresponding eigenvalues of $\boldsymbol{\Omega}$ are $\alpha = -0.102, -0.0444$. Both of these are negative and real, indicating that the fixpoint is a sink.

The stability diagram shown in **Figure 2B** in the main text was obtained in the plane of hydrodynamic length $\lambda$ and the active tension (or contractility) measure $\bar{\sigma} = \zeta' f(c_m^\star)$, where $c_m^\star$ denotes the stationary concentration of $c_m$ in the homogeneous state.

## Pulsatile dynamics of active RhoA

### Phenomenological description of the pulsatory dynamics

To investigate if an active RhoA pacemaking oscillator can control the contractile instability and determine the myosin pattern, we describe active RhoA pulsation dynamics by an oscillating standing wave. For this, we utilize a generic spatiotemporal oscillator, the complex Swift-Hohenberg equation (**Cross and Hohenberg, 1993**; **Sakaguchi, 1997**). We introduce a complex variable $\psi(x,t)$, the real part of which represents the concentration of active RhoA, $\Re[\psi(x,t)] = \rho_r(x,t)$. We write a conservation law for $\psi(x,t)$ according to

$$
\begin{aligned}
\partial_t \psi(x,t) &= -\partial_x j_r + R_{SH}(\psi(x,t) - \rho_{r0}(x,t)), \\
j_r &= \Re[\psi(x,t)] v + D_r \partial_x \Re[\psi(x,t)]
\end{aligned}
\tag{15}
$$

with

$$
\begin{aligned}
R_{SH}(\psi(x,t)) =\ & (a + if_0)\psi(x,t) - b(q_0^2 + \partial_x^2)^2 \psi(x,t) \\
& + if_1 \partial_x^2 \psi(x,t) - d_1 |\psi(x,t)|^2 \psi(x,t) \\
& - d_2 |\partial_x \psi(x,t)|^2 \psi(x,t).
\end{aligned}
\tag{16}
$$

Here, $R_{SH}$ is the pacemaking oscillator term using the complex Swift-Hohenberg equation, while $a, b, d_1, d_2, f_0, f_1$ and $q_0$ are phenomenological coefficients that determine the spatiotemporal dynamics of $\psi(x,t)$.

We chose a set of parameter values that is based on (**Sakaguchi, 1997**) and that gives rise to an oscillating pattern that is similar to the active RhoA oscillation shown in **Figure 3** of the

manuscript, with a characteristic length-scale of 5 $\mu m$ and a characteristic time-scale of 60 s ( i,e. the half-period corresponds to the active RhoA oscillation period). We find that the occurrence of a transition from a pacemaker- entrained state (i.e. the myosin pattern is determined by the pattern of the active RhoA pacemaker) to a detrained state (i.e. the active RhoA pattern is destroyed and now determined by cortical mechanics and myosin) is not sensitive to the detailed characteristics of the pacemaking oscillator, e.g. its amplitude, period, and spacing. For large hydrodynamic lengths $\lambda$, the myosin pattern is entrained to the active RhoA pulsatility. As we decrease $\lambda$, the period of myosin pattern oscillation increases, due to increasing amounts of accumulation of myosin and active RhoA at the myosin foci by advection. At intermediate values of $\lambda$ around $\sim 12$ $\mu m$, we observed a transition from the entrained state to the detrained state. Here, the dynamical pattern is characterized by irregular movements and 'mechanochemical turbulence' (**Figure 4** in main text, **Figure 4—figure supplement 1**), and depends on a both flow and advection as well as the active RhoA pacemaking activity.

## Associated time-scales

To provide an intuitive understanding of entrainment-detrainment transition, we compare characteristic time-scales of advection and biochemical regulations. First we obtain an advection time-scale, $\tau_{\mathrm{adv}}$, by solving **Equation 8** for $\partial_x f(\rho_{\mathrm{m}}) = \delta(x - x_0)$, where $\delta(x)$ is Dirac's delta function. For the infinite interval, $-\infty < x < \infty$, and a no-flux boundary condition, $v(\pm\infty, t) = 0$, the velocity profile, $v_0(x)$, is given by

$$v_0(x) = \frac{\zeta}{2\lambda\gamma}\exp\left(-\frac{|x - x_0|}{\lambda}\right). \tag{17}$$

hence, the velocity decay over space is characterized by the hydrodynamic length $\lambda$, while the magnitude of velocity is set by $V_c = \frac{\zeta}{2\lambda\gamma}$. We obtain a time-scale for advection as

$$\tau_{\mathrm{adv}} \sim \frac{\lambda}{V_c} = \frac{2\gamma\lambda^2}{\zeta}, \tag{18}$$

which specifies the time required for the thin film active fluid to advect biochemical regulators over the characteristic length $\lambda$.

Next, we introduce a characteristic time-scale of biochemical regulation as $\tau_{\mathrm{R}} = (\partial R_{\mathrm{m}}/\partial \rho_{\mathrm{r}})^{-1} = k_{\mathrm{on}}^{\mathrm{rm}-1} \sim 10$ $s$. If $\tau_{\mathrm{adv}} \gg \tau_{\mathrm{R}}$, the RhoA pacemaking oscillator controls the dynamics of the full mechanochemical patterning system, and advection plays a minor role. In this case, NMY-2 dynamics are entrained to the pulsatory dynamics of the active RhoApacemaker. In contrast to this, flow and advection dominate the dynamics of the full mechanochemical patterning system when $\tau_{\mathrm{adv}} \ll \tau_{\mathrm{r}}$. Here, the regularly oscillating pacemaker pattern of active RhoAis destroyed due to increased advection, leading to irregular spatiotemporal dynamics and 'mechanochemical turbulence'.

In the zygote, we find that for the non-RNAi case that $\tau_{\mathrm{adv}} = 16.5$ $s$ is larger than $\tau_{\mathrm{R}} \sim 10$ $s$. Thus the myosin pattern is entrained to the active RhoA pulsatory dynamics. On the other hand, we find that for the *pfn-1* RNAi case that $\tau_{\mathrm{adv}} = 6.5$ $s$, which is smaller than $\tau_{\mathrm{R}}$. Hence, we observe irregular behavior (**Figure 4B**, right, in the main text).

