## [Decision Letter]

Thank you for submitting your article "Controlling contractile instabilities in the actomyosin cortex" for consideration by *eLife*. Your article has been favorably evaluated by Marianne Bronner (Senior Editor) and three reviewers, one of whom is a member of our Board of Reviewing Editors. The following individuals involved in review of your submission have agreed to reveal their identity: Matthieu Piel (Reviewer #2); Jean-Francois Joanny (Reviewer #3).

The reviewers have discussed the reviews (see below) with one another and the Reviewing Editor has drafted this decision to help you prepare a revised submission.

In this article, Masatochi and colleagues combine quantitative imaging, advanced image analysis and modeling to study the formation of patterns in the acto-myosin cortex of a *C. elegans* zygote during the initial polarization event. They show that the observed Myosin II oscillatory patterns are driven by a RhoA oscillator that does not depend on the contractility of the cortex. They propose that this RhoA 'pacemaker' driving the acto-myosin patterns could be important to prevent large contractility occurring during polarization of whole cells, to lead to an instability of the cortex that would prevent it to exert the large forces needed, or to produce a cell-scale pattern. This is a very original and interesting idea.

As can be seen in the attached reviews, the reviewers have very different points of view on this paper. While all agree that the notion that contractile oscillations in the cortex "dampen" large instabilities is a new idea and the data analysis and modelling are novel and well done, reviewer 1 has concerns about the biological significance and the lack of new mechanistic insight into phenomenon of study, and reviewer 2 is not convinced, with the data provided, that the parameters measured in the experiments can really be used to fit the model.

The consensus of the reviewers, after discussion of their dissenting points of view, is that the manuscript could in principle be acceptable for publication in *eLife*, pending major revisions specifically addressing the following major points.

With regard to biological significance and/or mechanism, the authors would either need to provide some demonstration that RhoA oscillations mediating the damping of large contractile instabilities are a general phenomenon underlying other biological processes in which contractile pulses are known to occur, or provide some mechanistic dissection of the driver of Rho pulses. Other biological systems in which contractile pulses occur include several tissues in developing *Drosophila* as wells as *Xenopus* and mouse. Demonstration of myosin II-independent Rho activity oscillations in these systems would provide evidence of generality. The candidates for mechanistic insight into the Rho pulse driver/terminator include actin (see Nat Cell Biol. 2015; 17:1471-83. Bement WM. et al), or perhaps whatever lies between actin polymerization and Rho, likely some Rho-Gef. As noted by both reviewers, imaging and analysis of actin dynamics and application of drugs or mutants that stabilize the actin and thus inhibit its assembly/disassembly cycle are needed both to determine if the oscillations are driven by the same mechanism seen in the previous publication and to also better inform the parameters used in their model. The authors alternatively could screen known RhoGef mutants to determine their involvement in driving Rho oscillations.

With regard to better delineating the model parameters, results and interpretations, the authors need to rewrite the descriptions of the approaches, results and reasoning behind the conclusions more explicitly and in more detail.

*Reviewer 1:*

In this manuscript, the authors describe the regulation of actomyosin activity in the cortex of *C. elegans* zygotes. They use fluorescence microscopy, advanced image analysis, and theoretical modelling to try to understand the mechanism of the localized, pulsed assembly and movement of myosin 2 foci that occurs prior to and during establishment of embryonic polarity at the 1-cell stage. In the past several years, this pulsed activity of myosin 2 has been observed during developmental morphogenesis in worms, flies, frogs, and mice.

The authors show that the pulsed assembly of myosin 2 is driven by localized cortical pulses of RhoA activity, as reported by the Bement biosensor, and that RhoA pulses occur independently of myosin 2. Using previously characterized mutants, they show that the Rho kinase *let-502*, the centrosome-associated protein *spd-5*, and the profilin *pfn1* all regulate the ability of myosin 2 to pulse or form its normal pattern of pulsing, but have little effect on the pulsing of RhoA. The authors develop a theoretical model of the mechanochemical feedback between myosin 2 and RhoA based on a thin active gel, utilizing parameters for the temporal relationship between myosin 2 and RhoA pulses that they measure here, together with their previous measurements of cortical mechanical properties. This model predicts that in its wild type state, the zygote cortex verges just over the line between stability and instability. They predict that the rate of myosin 2 recruitment by RhoA and the hydrodynamic length of the cortex material determine the transition from stable and homogenous to unstable and pulsing. They show that perturbing the ability of RhoA pulses to recruit myosin 2 to the cortex with the *let-502* mutant abolishes myosin 2 pulses and makes the cortex homogeneous, while perturbing the integrity of the cortex with the *pfn-1* mutant induces greater instability and increases the dynamics of the myosin 2 pulses.

The phenomenon of pulsed or wave-like patterns of cortical activity has begun to be studied in depth. Previous studies by the Bement, Martin and Zallen labs have shown that Rho GTPases and their downstream effectors exhibit pulsed or wave-like dynamics in other embryonic systems, and that pulsing requires formin-mediated actin assembly, cortical integrity, and Rho-mediated myosin 2 phosphoregulation. The Bement lab has also demonstrated that Rho waves occur independent of myosin 2 activity, but their termination requires Rho-mediated actin assembly, thus placing actin at the hub of a mechanochemical negative feedback driving pulses. Whether actin assembly or some other mechanism is required for driving or terminating RhoA pulses in the *C. elegans* zygote was not explored in the current paper. Thus, the main contribution of the current study is the development of analysis methods and a theoretical model of cortical mechanics and dynamics that agrees with the notion that cortical integrity is a key driver of cytoskeletal pulses, and the finding from this model that the natural state of the cortical cytoskeleton in the *C. elegans* zygote verges on instability.

In spite of the flurry of studies of pulsed/wave-like cortical cytoskeleton behavior in the past several years, it is unclear whether the behavior is actually required for biological function, or is an epiphenomenon that occurs at a certain level of cortical contractility. The current paper suggests the latter. Indeed, they show here that *C. elegans* zygotes can undergo polarization in the absence of myosin 2 pulses, i.e. when the myosin 2 is homogeneously distributed in the cortex in the *let-502* mutant (although this mutant has other major defects in later stages of development). In addition, work from the Martin lab shows that in embryos that express a phosphomimetic myosin (with the endogenous myosin strongly reduced but not eliminated), apical constriction during mesoderm invagination in *Drosophila* occurs continuously (not in pulses), and yet the cells apically constrict and internalize just fine. Thus, the lack of advance into understanding the upstream regulation of the RhoA pulses together with the questionable biological significance of the pulses, unfortunately precludes publication of the paper in *eLife*.

*Reviewer 2:*

In this article, Masatoshi and colleagues combine quantitative imaging, advanced image analysis and modeling to study the formation of patterns in the acto-myosin cortex of a *C. elegans* zygote during the initial polarization event. They show that the observed Myosin II oscillatory patterns are driven by a RhoA oscillator that does not depend on the contractility of the cortex. They propose that this RhoA 'pacemaker' driving the acto-myosin patterns could be important to prevent large contractility occurring during polarization of whole cells, to lead to an instability of the cortex that would prevent it to exert the large forces needed, or to produce a cell-scale pattern. This is a very original and interesting idea. Experimentally estimated parameters are used in the frame of a mechanochemical model of the acto-myosin cortex which predicts an instability and formation of a spatial pattern. While the initial model proposed predicts the instability, it fails to predict the correct pattern and the pulsatile behavior. This behavior can be obtained by adding a driving pulsatile pattern in the modeling, which would correspond to the observed Myosin II independent pulsatile pattern of active RhoA. The origin of this RhoA pacemaker is not addressed in the paper, neither experimentally nor theoretically. Some perturbation experiments are performed to test predictions of the driven mechanochemical model. They suggest that the RhoA pacemaker might be essential to control the instability of the actomyosin cortex and form the observed pulsatile pattern instead of large clusters and flows. The final conclusion is new and represents an important step in the understanding of how the acto-myosin cortex works and contributes to morphogenesis. The experiments are well done and analysed, and there is no issue with the modeling proposed, which builds upon a large body of theoretical work and experiments – the modeling part was reviewed by my colleague Raphael Voituriez, who found no issue in the modeling proposed and commented that the part on the driving by the RhoA pacemaker, although leading to expected results, is new and well done. As a conclusion, the article should be published and will be an important contribution to the field.

I have a main concern, which might be solved mostly by a major effort in rewriting the article, if the authors are willing to do so. Indeed, my general feeling after many hours of work on this article, is that, despite all its qualities listed above, it does not convey its central message in a very convincing manner. The general feeling is that the authors tend to rush to the conclusions and leave the reader unsatisfied or feeling 'dumb' – asking the reader to be a believer rather than a critical thinker. The article is using a large number of sophisticated methods, from a number of different fields – developmental biology, imaging, physics – and if the authors want to be understood and trusted by scientists beyond the 'biophysicists of the acto-myosin cortex in *C. elegans*', they have to make a special effort in the presentation of their work.

*Reviewer 3:*

The manuscript of Nishikawa et al. studies both experimentally and theoreticallythe pulsatile contractile instability of the *C. elegans* actin cortex. The strength of this paper is the combination an original experimental technique to measure the kinetics of the actin cortex – that they call COMBI – to a theoretical analysis based on active gel theory. Each step of the theory is carefully checked experimentally and the authors get this way a very precise description of the cortical kinetics from which they describe the cortical instability. The instability is explained by an interplay between myosin molecular motors and the G protein RhoA. In order to explain the experimental results, the authors describe the dynamics of RhoA by introducing a RhoA pacemaker oscillator through the very generic Swift-Hohenberg equation.

This is a beautiful and very original manuscript and I certainly recommend its publication in *eLife*. The authors should consider the following comments prior to publication.

Throughout the manuscript the authors switch between RhoA and Rho1,which are if I understand well the same protein. It would be clearer for the reader if they were always using the same name.

The authors introduce a pacemaker dynamics of RhoA via the very general Swift-Hohenberg equation. This explains well the experimental results. However, they do not discuss the origin of this dynamics. Is there any information on that, which proteins are involved for instance?

The model proposed is only one-dimensional, it ignores the thickness of the cortex and restricts the description to one dimension along the cortex. It would of course be useful to extend the model to two dimensions to compare the observed patterns of each components with the prediction of the theory. That probably could be done numerically and would provide a stronger test of the theory.

One of the simplifications that the model makes is that it considers only an isotropic cortex. However, the active stress is not isotropic and depend on actin polarization. Does actin polarization play any role? What would it change to the theory?

One could also worry whether the instability described in this manuscript is generic. The main ingredient is the regulation of the myosin motors by Rho A, which to my knowledge is a very general feature and the pacemaking oscillations of RhoA. Would a similar instability be observed in any cortex?

---

## [Author Response]

*[…]*

*With regard to biological significance and/or mechanism, the authors would either need to provide some demonstration that RhoA oscillations mediating the damping of large contractile instabilities are a general phenomenon underlying other biological processes in which contractile pulses are known to occur, or provide some mechanistic dissection of the driver of Rho pulses. Other biological systems in which contractile pulses occur include several tissues in developing Drosophila as wells as Xenopus and mouse. Demonstration of myosin II-independent Rho activity oscillations in these systems would provide evidence of generality. The candidates for mechanistic insight into the Rho pulse driver/terminator include actin (see Nat Cell Biol. 2015; 17:1471-83. Bement WM. et al), or perhaps whatever lies between actin polymerization and Rho, likely some Rho-Gef. As noted by both reviewers, imaging and analysis of actin dynamics and application of drugs or mutants that stabilize the actin and thus inhibit its assembly/disassembly cycle are needed both to determine if the oscillations are driven by the same mechanism seen in the previous publication and to also better inform the parameters used in their model. The authors alternatively could screen known RhoGef mutants to determine their involvement in driving Rho oscillations.*

We thank the reviewer and the editor for these suggestions. The mechanistic basis for the RhoA pacemaking oscillator itself was not the main interest of our study. However, we agree that more information on the mechanism of pulse generation by RhoA would strengthen our work, and we addressed this issue in two ways.

First, we investigated the interaction between active RhoA and actin, by generating a transgenic line that expresses both the active RhoA probe and tagRFP-T labeled LifeAct, a probe for filamentous actin in cortex. We utilized this new line to determine a kinetic interaction diagram by using COMBI, as was done for the myosin and active RhoA in Figure 1 in the main text. As you can see in Figure 1—figure supplement 3, we did not detect negative feedback from actin onto RhoA, which is in contrast to what is seen by Bill Bement in the recent Nature Cell Biology paper (Figure 5C in Bement et al). Instead, the nullcline of RhoA kinetics (Figure 1—figure supplement 3, green line) displays a positive slope (i.e. active RhoA increases with increasing actin concentration), consistent with actin operating only as a positive regulator of active RhoA. This does not mean that there is no negative feedback at all between actin and RhoA, but suggests that the *C. elegans* cortex, in which there exists a dense network of actin, is normally operating in a regime where only positive feedback is accessed. To conclude, we do not observe quite as strong an interaction between actin and active RhoA in *C. elegans* as was reported for *Xenopus*.

Second, following the advice from the editor we set out to determine if known RhoGEFs are involved in driving the RhoA pacemaking oscillator. Previous work has shown that the *C. elegans* RhoGEF ECT-2 positively regulates RHO-1 (see e.g. Schonegg and Hyman, 2006), and we now demonstrate that RNAi of this RhoGEF leads to the absence of RhoA pulsation (Video 6). This allows us to identify ECT-2 as a positive regulator of RhoA, even if the kinetics aren’t clear yet. This is now properly discussed in the revised manuscript.

While these additional experiments strengthen our work, we hope that the referees and the editor agree that an analysis of the full kinetics of recruitment and activation underlying active RhoA pulsation, which is required for understanding how they arise, is beyond the scope of the work here. This is the focus of ongoing research and (hopefully) a future publication from our lab.

*With regard to better delineating the model parameters, results and interpretations, the authors need to rewrite the descriptions of the approaches, results and reasoning behind the conclusions more explicitly and in more detail.*

We agree and have added better descriptions throughout. We have also taken most of the criticism raised below into account in our revised manuscript. Please also note our responses to the detailed concerns given below.

*Reviewer 1:*

*[…] The phenomenon of pulsed or wave-like patterns of cortical activity has begun to be studied in depth. Previous studies by the Bement, Martin and Zallen labs have shown that Rho GTPases and their downstream effectors exhibit pulsed or wave-like dynamics in other embryonic systems, and that pulsing requires formin-mediated actin assembly, cortical integrity, and Rho-mediated myosin 2 phosphoregulation. The Bement lab has also demonstrated that Rho waves occur independent of myosin 2 activity, but their termination requires Rho-mediated actin assembly, thus placing actin at the hub of a mechanochemical negative feedback driving pulses. Whether actin assembly or some other mechanism is required for driving or terminating RhoA pulses in the C. elegans zygote was not explored in the current paper. Thus, the main contribution of the current study is the development of analysis methods and a theoretical model of cortical mechanics and dynamics that agrees with the notion that cortical integrity is a key driver of cytoskeletal pulses, and the finding from this model that the natural state of the cortical cytoskeleton in the C. elegans zygote verges on instability.*

Indeed, the focus of our work is the investigation of cortical integrity, and the mechanisms by which a contractile instability can be controlled. An in-depth investigation of the mechanisms by which the RhoA oscillator is generated was not the core focus of our study, since this has been worked on before. However, this referee, together with referee #2, requested additional information on which mechanisms contribute to positive and/or negative feedback in RhoA. We have taken these comments seriously, and have added two pieces of additional data. First, we performed a kinetic COMBI analysis of active RhoA vs. actin, and find that in contrast to *Xenopus*, actin does not appear to be contributing to the termination of RhoA pulses. Second, we show that RhoA pulses require the Rho-GEF ECT-2 (see our answer above for more detail). Together, these results shed further light on the RhoA oscillator, which in turn, as we show here, provides increased cortical integrity by controlling the contractile instability.

*In spite of the flurry of studies of pulsed/wave-like cortical cytoskeleton behavior in the past several years, it is unclear whether the behavior is actually required for biological function, or is an epiphenomenon that occurs at a certain level of cortical contractility. The current paper suggests the latter. Indeed, they show here that C. elegans zygotes can undergo polarization in the absence of myosin 2 pulses, i.e. when the myosin 2 is homogeneously distributed in the cortex in the let-502 mutant (although this mutant has other major defects in later stages of development). In addition, work from the Martin lab shows that in embryos that express a phosphomimetic myosin (with the endogenous myosin strongly reduced but not eliminated), apical constriction during mesoderm invagination in Drosophila occurs continuously (not in pulses), and yet the cells apically constrict and internalize just fine. Thus, the lack of advance into understanding the upstream regulation of the RhoA pulses together with the questionable biological significance of the pulses, unfortunately precludes publication of the paper in eLife.*

Any contractile medium is prone to contractile instabilities, this is one of the points we are making in our work. If the material is too contractile, it will tend to pull itself apart, this is inevitable. Thus, one might refer to this behaviour as an epiphenomenon, however, we don’t think that this terminology does justice to the problem at hand. Controlling this instability allows the material to remain highly contractile overall to successfully, rapidly and robustly drive morphogenetic events, this is our conclusion. Submitting to the contractile instability gives rise to a lethal phenotype, as we show via pfn-1 RNAi. Hence, our work assigns a potential function to the RhoA/myosin pulses, which is to control the instability and allow the cortex to operate in a regime of high contractility that is otherwise not accessible. Removing pulsation does not appear to generally give rise to lethal phenotypes, this is correct. However, as stated above our suggestion is that overall contractility will necessarily be lower, leading to less rapid and probably less robust morphogenetic transitions. This is now more explicitly discussed in our revised manuscript. We hope that the referee now agrees that our work provides a significant step forward both for the biology and the physical mechanism of myosin pulsation.

*Reviewer 2:*

*[…] I have a main concern, which might be solved mostly by a major effort in rewriting the article, if the authors are willing to do so. Indeed, my general feeling after many hours of work on this article, is that, despite all its qualities listed above, it does not convey its central message in a very convincing manner. The general feeling is that the authors tend to rush to the conclusions and leave the reader unsatisfied or feeling 'dumb' – asking the reader to be a believer rather than a critical thinker. The article is using a large number of sophisticated methods, from a number of different fields – developmental biology, imaging, physics – and if the authors want to be understood and trusted by scientists beyond the 'biophysicists of the acto-myosin cortex in C. elegans', they have to make a special effort in the presentation of their work.*

We thank the referee for this feedback, which we have taken seriously in our re- vision.

*Reviewer 3: […] Throughout the manuscript the authors switch between RhoA and Rho1,which are if I understand well the same protein. It would be clearer for the reader if they were always using the same name.*

This has been changed.

*The authors introduce a pacemaker dynamics of RhoA via the very general Swift-Hohenberg equation. This explains well the experimental results. However, they do not discuss the origin of this dynamics. Is there any information on that, which proteins are involved for instance?*

This concern resonates well with similar concerns from the other referees. We have done additional experiments and now report on the involvement of Rho GEF, *ect-2* in generating the oscillatory dynamics of the active RhoA. Please see the response to the editor above for detail.

*The model proposed is only one-dimensional, it ignores the thickness of the cortex and restricts the description to one dimension along the cortex. It would of course be useful to extend the model to two dimensions to compare the observed patterns of each components with the prediction of the theory. That probably could be done numerically and would provide a stronger test of the theory.*

A precise comparison between 2D foci dynamics with the RhoA oscillator in place in theory and simulation is beyond what we are aiming to do in our manuscript. We have put in place a generic spatiotemporal oscillator for the RhoA dynamics that will differ from the actual dynamics, and consequently we are not expecting that the dynamics of the full system in 2D will be quantitatively reproduced with our approach here. Indeed, we had and have done 2D simulations and they show similar general behavior. Given that a 2D analysis might raise false hope that the 2D RHoA-myosin patterns are quantitatively reproduced in theory (which they are not given the uncertainties underlying the nature of the RhoA oscillator, neither for 1D nor for 2D), we have chosen not to add this to the manuscript. We have also added a clarification that our theory remains qualitative and does not quantitatively reproduce the patterns.

*One of the simplifications that the model makes is that it considers only an isotropic cortex. However, the active stress is not isotropic and depend on actin polarization. Does actin polarization play any role? What would it change to the theory?*

This is a very good point that also relates to the point just above (2D simulation). We have considered isotropic active stresses only, which we know is a simplification (see our recent *eLife* paper Reymann et al.). Introducing a coupling between active tension and alignment would alter the theory, however, we think this will be an effect that will mildly impact the dynamics in a 2D simulation. This type of comparison is beyond what we are aiming to do here and we have chosen to remain qualitative and have not added 2D simulations (see our answer to the point above). Finally, moving from nematic to polar order will again change the dynamics, but we have no experimental access to the local state of polarization of the cortex so this is currently not really feasible.

*One could also worry whether the instability described in this manuscript is generic. The main ingredient is the regulation of the myosin motors by Rho A, which to my knowledge is a very general feature and the pacemaking oscillations of RhoA. Would a similar instability be observed in any cortex?*

We think the instability is generic. First of all, any cortex that is overly contractile will undergo a contractile instability. Perhaps the referee is more concerned with the actual feedback we are reveling for *C. elegans*, which is more system-specific. While the kinetics will be different in other systems, RhoA is a general master-regulator of the cortex, and we expect a similar feedback to exist in essentially every animal cortex. And the ability to cross the contractile instability critically depends on the level of mechanochemical feedback and myosin recruitment, and as such the mechanism should be generic.